# Research Progress on the Collaborative Drag Reduction Effect of Polymers and Surfactants

**DOI:** 10.3390/ma13020444

**Published:** 2020-01-17

**Authors:** Yunqing Gu, Songwei Yu, Jiegang Mou, Denghao Wu, Shuihua Zheng

**Affiliations:** 1School of Measurement and Testing Engineering, China Jiliang University, Hangzhou 310018, China; mjg@cjlu.edu.cn (J.M.); wdh@cjlut.edu.cn (D.W.); 2College of Mechanical Engineering, Zhejiang University of Technology, Hangzhou 310023, China; yu752115892@163.com (S.Y.); zneu@zjut.edu.cn (S.Z.)

**Keywords:** Drag reduction agent, polymer, surfactant, collaborative drag reduction, anti-shearing, complexes

## Abstract

Polymer additives and surfactants as drag reduction agents have been widely used in the field of fluid drag reduction. Polymer additives can reduce drag effectively with only a small amount, but they degrade easily. Surfactants have an anti-degradation ability. This paper categorizes the mechanism of drag reducing agents and the influencing factors of drag reduction characteristics. The factors affecting the degradation of polymer additives and the anti-degradation properties of surfactants are discussed. A mixture of polymer additive and surfactant has the characteristics of high shear resistance, a lower critical micelle concentration (CMC), and a good drag reduction effect at higher Reynolds numbers. Therefore, this paper focuses more on a drag reducing agent mixed with a polymer and a surfactant, including the mechanism model, drag reduction characteristics, and anti-degradation ability.

## 1. Introduction

At present, many countries in the world have laid pipe network systems to transport petroleum, natural gas, and water. Friction resistance is the main form of fluid energy loss in pipeline transportation. Consequently, a large number of pumping stations must be built along the line to provide additional energy to overcome fluid resistance. One aspect of the important data on pipeline transportation is transportation capacity. However, factors used to estimate conveying capacity are still uncertain. Therefore, the selection of pipe diameter, pump station, and equipment is conservative, which may lead to construction investment costs that are higher than those of the actual operation.

To reduce resistance loss in the conveying flow, a drag reduction method is often used. Drag reduction methods can be generally divided into two categories, namely non-additive drag reduction method [1] and additive drag reduction method [2]. Non-additive drag reduction method includes the coating drag reduction method [3], the non-smooth surface drag reduction method [4], etc. The additive method injects a polymer or surfactant into the pipeline to achieve drag reduction. This method was first proposed by Toms in 1949. He found that adding polymers to low viscosity media can reduce the flow resistance of fluids. Since then, researchers have studied the drag reduction of additives [5]. Adding a drag reducing agent into the pipeline can reduce the pressure drop and the number of pump stations, thereby reducing construction costs. This drag reduction technology was applied to the Alaskan oil pipeline project as early as 1982, and has achieved remarkable results. When the drag reducer is injected into the pipe, the polymer molecules fully expand and interfere with the flow from the laminar region to the turbulent layer.

The additive drag reduction method can change the pipeline capacity from rigid to flexible, reduce the safety risk, save the investment costs of the pipeline and pump station, and reduce the operational and maintenance costs. Overall, the reasonable use of a drag reducer is more economical. With the development of research on drag reducers by researchers all over the world, the additive drag reduction method can achieve a maximum drag reduction rate of about 80% [6]. The drag reduction agent method has the advantages of a strong drag reduction effect and low investment costs. It is widely used in oil and gas transportation, fire protection engineering [7], irrigation water, regional heat transfer or cooling [8], cardiovascular disease [9], and other fields.

However, polymer drag reducers have inherent limitations. The polymer’s long chain undergoes irreversible mechanical degradation under shear. A surfactant has anti-degradation properties. However, to obtain the drag reduction effect, the surfactant concentration must reach the critical micelle concentration (CMC), and its chemical properties may have adverse effects on the wall and solvent. Therefore, some researchers have focused on a new type of drag reducer that is a mixture of the two. This paper presents the current drag reduction mechanism categories and the degradation characteristics of drag reduction agents. Then, the model mechanism, drag reduction and anti-degradation characteristics of the mixed drag reducer are reviewed, and the prospects for development are given at the end of the paper.

## 2. Development of the Drag Reduction Mechanism of a Drag Reducer

From the physical point of view of turbulence, the mechanism of turbulent drag reduction is applicable to both polymer-type drag reducers and surfactant-type drag reducers. Therefore, in the following description of this aspect, there is no distinction made between the types of chemical drag reducers, although some explanations for the mechanism of turbulent drag reduction are based on polymers. However, when explaining the drag reduction mechanism from the perspective of the internal microstructure of a drag reducer, the microstructure of the two is different. Therefore, special elaboration is required.

### 2.1. Pseudoplasticity Hypothesis

Research on the drag reduction mechanism of a drag reducer began in 1948. After Toms discovered the drag reduction phenomenon, he put forward the pseudoplasticity hypothesis and thought that the polymer drag reduction agent solution had pseudoplasticity [5]. Later, with the development of non-Newtonian fluid mechanics, it was revealed that the friction resistance of the drag reducer solution in the turbulent flow in the pipe is quite different from the calculated value of the pseudoplastic fluid, and the pseudoplasticity of the dilute solution is very weak, or even does not exist at all. This fundamentally negates the pseudoplasticity hypothesis.

### 2.2. Effective Slip Hypothesis

Virk proposed the effective slip hypothesis from the turbulence structure [10]. It is assumed that the flow field can be divided into three layers according to the velocity distribution. From the outside to the inside, they are the viscous sublayer, the elastic layer, and the turbulent core layer. By measuring the flow velocity of different solutions and drag reducers, the curve of drag reducers under different working conditions is obtained. Figure 1 is a schematic diagram of the stratification of turbulence in the pipe flow [11]. As shown in Figure 1, the injection of drag reducers weakens the pulsation and inhibits the formation and development of vortices.

According to his hypothesis, drag reducers act on the elastic layer. The elastic layer expands with the increase in drag reducer concentration, and the drag reduction rate reaches a maximum value when it extends to the center of the tube axis. This hypothesis can explain the diameter effects, but it does not conform to the basic principle of the continuity of the velocity, and cannot explain the resistance rising when the concentration continues to increase after the drag reduction rate reaches the maximum value [12], nor the phenomenon of the change of thermal conductivity coefficient [13,14]. Therefore, it was gradually replaced by other hypotheses.

### 2.3. Turbulence Suppression Hypothesis

With the emergence of advanced measuring instruments, researchers found that the laminar flow close to the pipe wall has not only axial velocity fluctuation, but also velocity fluctuation in the direction perpendicular to the pipe wall [15]. When the laminar flow changes to turbulence, it is often accompanied by the generation of the vortex, resulting in a huge energy loss. When the drag reducer is added, the fluctuation of the turbulent bottom layer near the wall is restrained by the long-chain structure of the polymer, which makes the fluctuation decrease. The downstream Reynolds shear stress and wall-normal velocity fluctuation decrease. Therefore, it is considered that drag reduction is achieved by reducing turbulence intensity. This may seem reasonable at first, but it is not consistent with the measured results of turbulence fluctuation intensity, because the effect of drag reducers is not only to reduce the intensity of turbulence, but also to change the vortex structure of turbulence. Downstream of the area where the drag reducer is added, the hairpin vortex along the shear layer is significantly reduced, and the energy loss caused by the corresponding vortex is reduced [16,17].

Later, some researchers explained the mechanism of drag reduction from the perspective of turbulence fluctuation decoupling, and further divided the fluid with a drag reducer into many layers on the micro-level, believing that the generation of the vortex between each layer of fluid was inhibited [18]. Hidema [19] added poly(ethylene) oxide (PEO) as a flexible polymer and hydroxypropyl cellulose (HPC) as a rigid polymer in the turbulent flow field. The vortex structure of the flow field shown in Figure 2 was obtained through the interferogram and particle image velocimeter. It is observed that the vortices in the flow shown in Figure 2b become long and thin. The polymer stabilizes the primary vortex from breaking. However, it is difficult to predict the effect of drag reduction by quantitative evaluation.

### 2.4. Viscoelasticity Hypothesis

The viscoelasticity hypothesis holds that the fluid with a drag reducer has viscoelasticity [20,21,22]. Polymer molecules form a long-chain molecular structure and a spiral structure [23], and the surfactant can form a micelle structure [24], as shown in Figure 3. These structures can absorb part of the kinetic energy of the vortex in the turbulent process. The kinetic energy is absorbed and stored in the form of elastic properties. This leads to a decrease in eddy current friction loss and explains the reason from the perspective of energy. Viscoelasticity is caused by the entanglement or association of molecular chains, so the solution of high molecular weight polymers shows an obvious drag reduction effect at a lower concentration, which is the advantage compared with the surfactant. The micelle structure of the surfactant recovers after degradation, but the degradation of the polymer is irreversible.

Surfactants gradually aggregate into spherical, rodlike, and wormlike micelles, and form a network structure under flow shear. The wormlike structure is easily affected by water flow and parallel to the water flow’s direction. This microstructure results in a large critical Reynolds number and drag reduction effect. Some wormlike structures also form bifurcations [25]. The bifurcation point is not fixed; it can move freely along the axial direction of the micelle structure, thus reducing the shear stress and the shear viscosity of the solution. Some surfactant solutions form vesicles when they are still. When the shear rate exceeds the critical value, the solution can be transformed into a wormlike structure. The surfactant’s microstructure is related to the drag reduction effect. When the internal microstructure of the solution is destroyed, the turbulence structure and turbulence velocity fluctuation cannot be changed, and the effect of turbulent drag reduction is lost.

As a special turbulent phenomenon, the drag reduction effect is a complex result of the drag reducer’s influence on the flow field’s microstructure. The molecular structure and turbulence structure of drag reducers have some limitations for the study of drag reduction mechanism. After adding the drag reducing agent, the vortex size and the fluctuation decrease on the macroscale. However, to explain the causes of turbulence structure changes, it is necessary to focus on the microscale and the movement of the drag reducer.

## 3. Mechanical Degradation of a Drag Reducer

At present, the main drag reducers are polymers and surfactants. A polymer drag reducer is sheared mechanically when it flows through the elbow, valve, pump, and other flow parts in the pipeline. Degradation also occurs when it is heated, and this mechanical degradation is irreversible.

### 3.1. Drag Reduction Is Related to Molecular Weight and Molecular Weight Distribution

Quan [26] studied the relationship between the drag reduction rate of a polymer drag reducer and the time when it was sheared in an organic solution. The results showed that with the increase in shear time, the drag reduction rate decreased linearly and finally approached zero. However, polymer drag reducers are still widely used because only a small amount of a polymer drag reducer can achieve significant pressure drop and drag reduction effect. Polymer drag reducers degrade, so it is necessary to supplement drag reducers regularly.

The mechanical degradation of polymer drag reducers in turbulent flow depends on many parameters. Brostow [27,28] studied the relationship between degradation and polymer molecular weight change in an organic solution and proposed an important formula, that is, drag reduction efficiency is directly proportional to the molecular weight of polymer, as shown in Equation (1). *DR*(*t*) and *DR*(0) are the drag reduction rate before and after degradation, and *M*(*t*) and *M*(0) are the molecular weight before and after degradation.
(1)$$\frac{DR\left(t\right)}{DR\left(0\right)}=\frac{M\left(t\right)}{M\left(0\right)}$$

However, Liberatore et al. [29] studied the drag reduction effect of a polyacrylamide (PAM) aqueous solution with different concentrations in the circulating flow. As shown in Figure 4, when the drag reduction rate dropped from 58% to 42% in the initial stage, the molecular weight and molecular weight distribution basically remained unchanged, and the drag reduction rate decreased with the circulation time, proving that the change of drag reduction rate was not in direct proportion to the change of molecular weight.

Zhang et al. [30] also verified Brostow’s hypothesis through experiments. They conducted drag reduction experiments on three different molecular weight PEO solutions. Based on the first-order chemical reaction of PEO degradation, the correlation of the Arrhenius equation was improved, and the correlation error of the drag reduction rate prediction was ±15%. The drag reduction effect is related to molecular weight, molecular weight distribution, and interactions between polymer chains. It is difficult to predict the impact of degradation on drag reduction, and most prediction equations apply to limited drag reducers.

### 3.2. Factors Affecting the Mechanical Degradation of Polymer Drag Reducers

Temperature, polymer concentration, Reynolds number, and other parameters are related to the degradation of a polymer drag reducer [31,32,33]. Hasan et al. [34] studied the effect of temperature on the degradation rate of critical micelle concentration (CMC). The concentration of CMC solution decreased by 17.7% at 37.8 °C when it was rotated for 1 h at the same shear rate of 100 revolutions per minute (RPM). The concentration of CMC solution at 104.4 °C decreased by 36.2%. It is proved that the degradation rate of the polymer synthesized at high temperature is faster. Habibpour and Clark [35] studied hydrolyzed polyacrylamide solutions of different concentrations. It was found that with the increase in hydrolyzed polyacrylamide concentration, the shear degradation resistance of the solution increased. Yang and Ding [36] used the concept of elastic shear stress or Reynolds shear stress deficit to derive the velocity distribution and friction coefficient equations in turbulent drag reduction flow. The initial Reynolds number of drag reduction depends on the type of polymer and its concentration. The optimum polymer concentration for the maximum drag reduction ratio depends only on the type of polymer.

### 3.3. Mechanical Degradation and Self-Assembly Characteristics of Surfactant Drag Reducer

Surfactant drag reducers can form micelle structures with self-recovery ability [37]. The results in [38] show that the micellar system has strong viscoelasticity, shear-thinning characteristics, and network structure. The shear viscosity of fluid changes with shear rate through three stages—shear thinning, shear thickening, and secondary shear thinning. When the drag reducing fluid is sheared, the micelles in the state of relaxation are close to each other, twined and even connected (forming a shear-induced structure) under the shear action, resulting in the viscosity of the solution increasing gradually. As the shear continues, the micelle structure is formed accompanied by destruction, so the micelle structure of the surfactant keeps a dynamic balance, which is called self-assembly characteristics [39]. In the process of self-assembly, the micelle morphology changes dynamically. Yan et al. [40] studied the self-assembly behavior of amino acid-based anionic N-hexadecanoylglutamic acid (HGA) and cationic benzyldimethyl hexadecylammonium chloride (HDBAC) surfactant in water, and found that the micelle morphology changed with the increase in HDBAC concentration. The formation and evolution of spherical micelles are shown in Figure 5a, while the gradual transformation of spherical micelles into wormlike micelles is shown in Figure 5b–d. The morphology changes from spherical micelle to wormlike micelle. A certain concentration of surfactant is needed to form the best micelle structure, called the critical aggregation concentration (CAC). Only when the concentration reaches CAC can the turbulent drag reduction effect appear.

## 4. Characteristics of Surfactant–Polymer Complexes

Through an in-depth study of the complex formed by the interaction between surfactant and polymer, it was found that adding a surfactant to a polymer solution can change the degradation characteristics of a drag reducer. Shear resistance is an important index for evaluating a drag reducer’s performance. Further optimization of a drag reducer’s drag reduction effect involves rheology, hydrodynamics, molecular dynamics, and other disciplines. At present, it is a breakthrough point to study the drag reduction characteristics of the common surfactant and polymer reactants to improve the limitations of traditional drag reduction agents.

### 4.1. Theoretical Models of Complexes

Shirahama et al. [41] first used the electrophoretic method to study the complex of the surfactant and polymer interaction, and proposed the bead model as shown in Figure 6. Surfactant molecules adsorb (bind) onto the polymer’s molecular chain in the form of micelle-like aggregates, resulting in a synergistic effect. Later researchers also proposed a “sphere” model [42] and a “three-stage” model [43] to explain the mechanism of the mixtures’ interaction.

The drag reduction characteristics of the surfactant–polymer mixture under different pH conditions were also studied. Kim et al. [44] added a surfactant, sodium dodecyl sulfate (SDS), to a polyacrylic acid (PAA) aqueous solution to study the surfactant–polymer complex at different pH levels. It was found that the molecular bond forces of the polymer molecules resulted in a chain expansion. The extended polymer conformation is shown in Figure 7. The results show that the highly extended structure is more beneficial to drag reduction than the compact spiral structure.

Liu et al. [45] studied the drag reduction effect of hexadecyltrimethylammonium chloride (CTAC) and a PAM mixed aqueous solution, and proposed the complex network model as shown in Figure 8. It was considered that polymer molecules act as reinforcement and surfactant micelles act as concrete. The micelle structure formed by the surfactant surrounds the polymer, thus protecting the polymer’s molecular chain from shear stress. The big net is the polymer frame, the small net is the surfactant, and the big net is filled by the small net. The frame improves its stability by winding large and small nets. The network structure model with an intensive internal cross-linking and stable structure is used to explain the synergistic effect.

These results show that most of the molecular chains of high-efficiency drag reducers have a flexible linear or spiral structure, and their molecular chains have some branches. Grafting a short branch on the main chain of a polymer molecule reduces the drag reduction performance, whereas grafting a small number of long branches enhances the drag reduction effect. The length, number, and distribution of branches are related to the drag reduction rate. There are two types of drag reducers—one is the surfactant grafted onto the long chain of the polymer, the other is the polymer surrounded by the micelle formed by the surfactant. Therefore, the drag reduction effect of the complexes formed by polymers and surfactants is quite different.

### 4.2. Drag Reduction Properties of Complexes

When a surfactant is added to a polymer solution, the critical aggregation concentration, or CAC, of the surfactant molecule is affected by the polymer, which is usually smaller than the critical micelle concentration, or CMC. Suksamranchit et al. [46] studied the drag reduction performance of a mixture aqueous solution containing a cationic surfactant, cetyltrimethylammonium chloride (HTAC), and a nonionic polymer, polyethylene oxide (PEO). It was found that the wall shear stress of the diluted HTAC solution that was lower than the CMC also significantly decreased with the concentration of HTAC. Compared with a single surfactant, the addition of polymers can improve the ability of the surfactant to form micellar structures that are lower than the CMC.

Matras and Kopiczak [47] added sodium salicylate to an aqueous solution of the HTAC and PEO complex as a counterion. The results showed that the complex still had a high drag reduction ratio in a wider range of Reynolds numbers. The counterion increased the critical shear stress, which proves that the addition of a counterion to a mixed solvent of surfactant and polymer can further enhance the drag reduction effect. Compared with the drag reduction effect of a pure polymer or surfactant, the effective drag reduction range of the complex is significantly expanded.

The complex reacts through the hydrophobic interaction of a surfactant and nonionic polymers. Its characteristic is that the CAC of the complex is close to that of the single surfactant. In addition, the mixed reaction must add a counterion such as sodium salicylate to achieve the effect of high drag reduction. This occurs because the counterion neutralizes the charge and make the micelle easier to form. Furthermore, the type of surfactant and the addition of long-chain alcohols, metal ions, and anti-ions affect the drag reduction properties of the complex. At a high Reynolds number, the polymer’s drag reduction effect is greatly weakened, whereas the complex has better structural stability and degradation reversibility similar to the micelle structure, giving the complex a larger effective drag reduction range.

### 4.3. Anti-Degradation Properties of Complexes

Guilherme et al. [48] combined xanthan gum, polyacrylamide, and poly(ethylene oxide) with guar gum, respectively, and found that the mixture aqueous solution of guar gum and polyacrylamide at a ratio of 1:4 had a greater resistance to shear degradation. The results showed a good complexing reaction between the rigid-chain polymer and the flexible-chain polymer. Salleh et al. [49] studied the anti-degradation ability of the polyisobutylene and sodium lauryl ether sulfate complex in an organic solution. The results showed that the maximum drag reduction rate observed in the laminar flow at 1000 rpm was 38.42%. Under the same conditions, the maximum drag reduction rates of polyisobutylene and sodium lauryl ether sulfate were 27% and 28.42%, respectively. Ferhat et al. [50] added a surfactant to a polymer aqueous solution and carried out degradation tests at 25 °C to 50 °C. It was found that the drag reduction effect of the pure polymer solution without surfactant was significantly reduced at high temperature, whereas that of the mixed solution with surfactant was not significantly reduced at high temperature.

These studies show that adding a surfactant to a polymer solution may be an effective way to reduce the mechanical degradation of the polymer, especially in a high-temperature flow system. The thermal degradation resistance of the polymer is significantly improved by adding the surfactant. In some cases, the drag reduction performance of the associated polymer is better than that of the pure polymer alone, which is more obvious in turbulence with a higher Reynolds number.

## 5. Conclusions

As an efficient drag reduction technology, additive drag reduction technology has great application prospects because of its economic and reliable advantages. However, this technology still has some shortcomings, such as the sensitivity of the drag reduction polymer to mechanical and thermal degradation, the concentration conditions needed for the surfactant to play a drag reduction role, and so on. These characteristics limit the application of drag reducers. However, the new drag reducer formed by a surfactant–polymer mixture has the advantages of strong shear resistance, low concentration, and high drag reduction rate at a higher Reynolds number, and the micelle structure’s critical concentration is reduced. The method of adding the surfactant to the polymer solution to improve the shear resistance and drag reduction effect of a mixed drag reducer is simple and easy to operate without further preparation.

Combining the advantages of a polymer and a surfactant, research on a new drag reducer is a development direction for the future. Modern testing methods must be used in future work to study the influence of complexes on a convection field’s structure from a macro perspective. The composition, structure, and interaction mechanism of complexes have been studied from a microscopic point of view, which can provide guidance for obtaining a mixture with high-efficiency drag reduction and reducing blind attempts. Future work must study how this complex structure can play a synergistic role in drag reduction under turbulent conditions and how to select reaction parameters to obtain a high-efficiency polymer–surfactant hybrid drag reducer.

## Figures and Tables

**Figure 1 materials-13-00444-f001:**
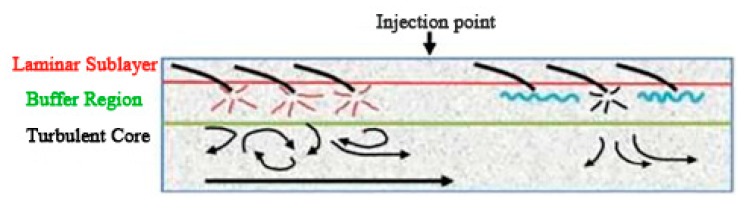
Schematic diagram of turbulent flow in the pipeline [11].

**Figure 2 materials-13-00444-f002:**
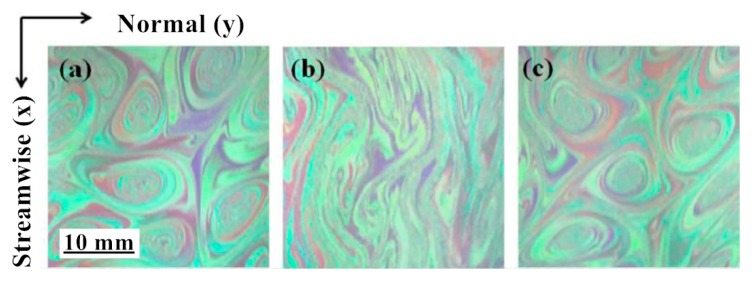
Two-dimensional turbulent stratification with polymer: (**a**) polymer-free solution; (**b**) poly(ethylene) oxide (PEO) 1.5 × 10^−3^ weight percentage solution; (**c**) hydroxypropyl cellulose (HPC) 0.08 weight percentage solution [19].

**Figure 3 materials-13-00444-f003:**
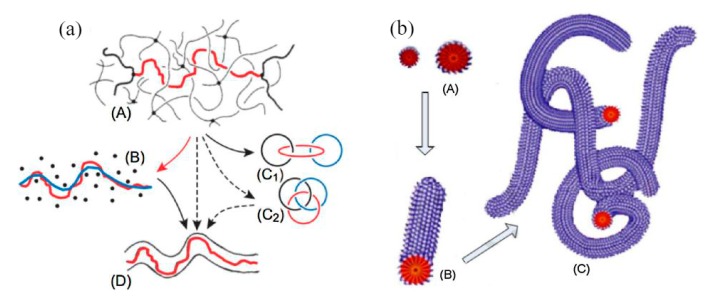
Microstructure of polymer drag reducer and surfactant in the fluid: (**a**) high-molecular polymer solution [23]; (**b**) surfactant solution [24].

**Figure 4 materials-13-00444-f004:**
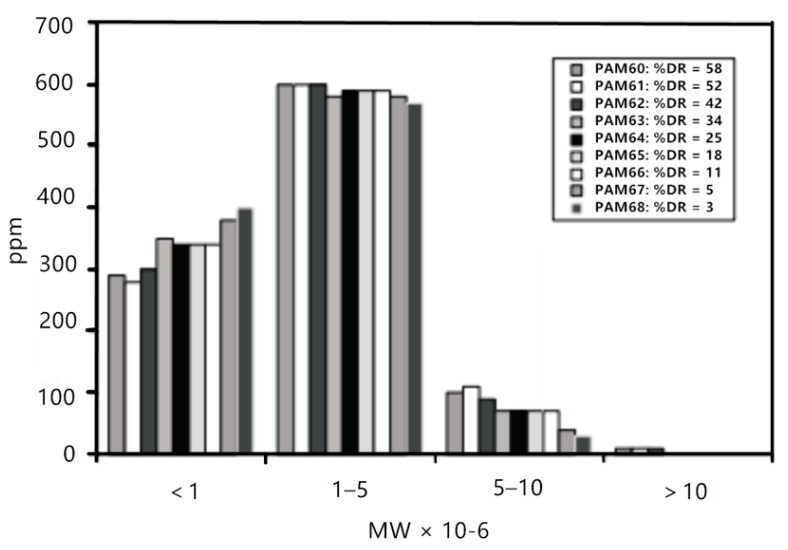
Molecular weight distribution for 1000 ppm polyacrylamide (PAM) solutions from channel flow experiments under drag reduction conditions [29].

**Figure 5 materials-13-00444-f005:**
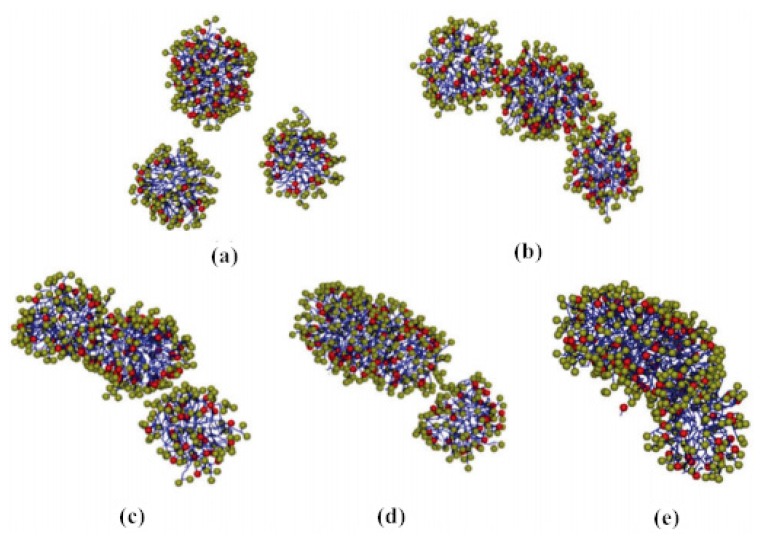
Transition process from globular micelle to wormlike micelle: (**a**) dispersed spherical micelles; (**b**) spherical micelles approach each others; (**c**) spherical micelles begin to merge; (**d**) two spherical micelles merge first; (**e**) three spherical micelles combine to form wormlike micelles [40].

**Figure 6 materials-13-00444-f006:**
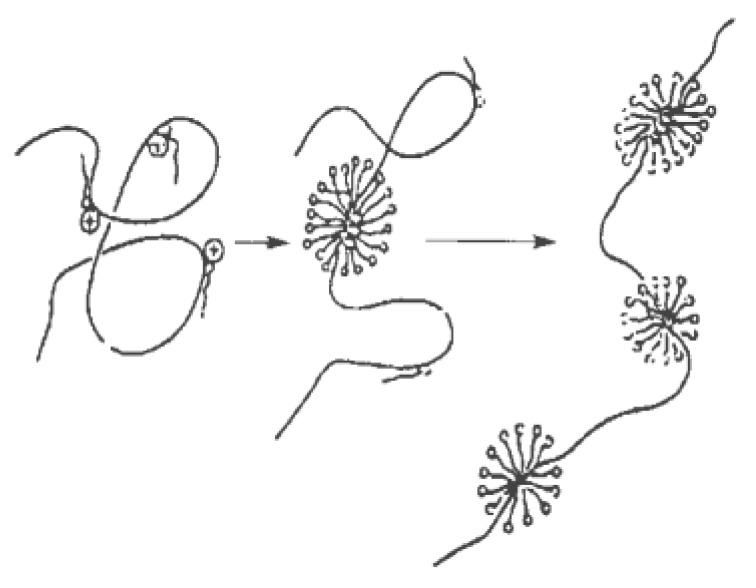
Bead model of the surfactant–polymer mixture [41].

**Figure 7 materials-13-00444-f007:**
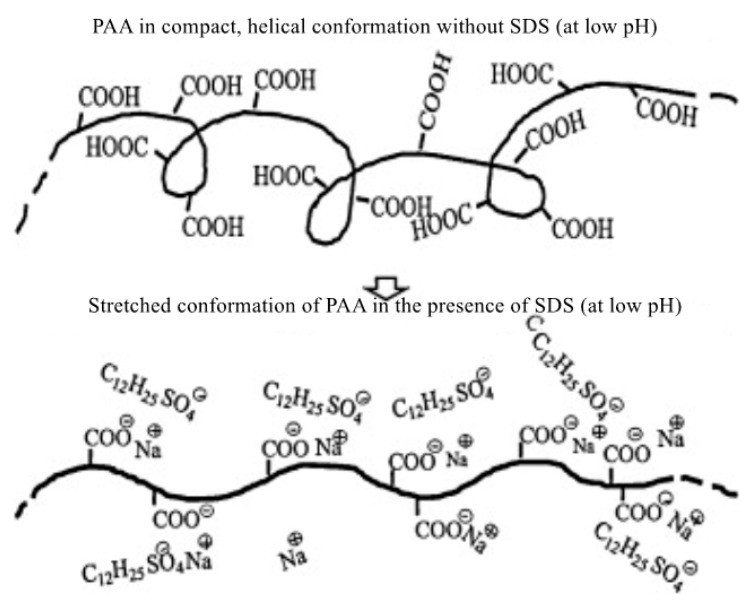
Schematic diagram of the conformational change of the PAA molecule at different pH levels [44].

**Figure 8 materials-13-00444-f008:**
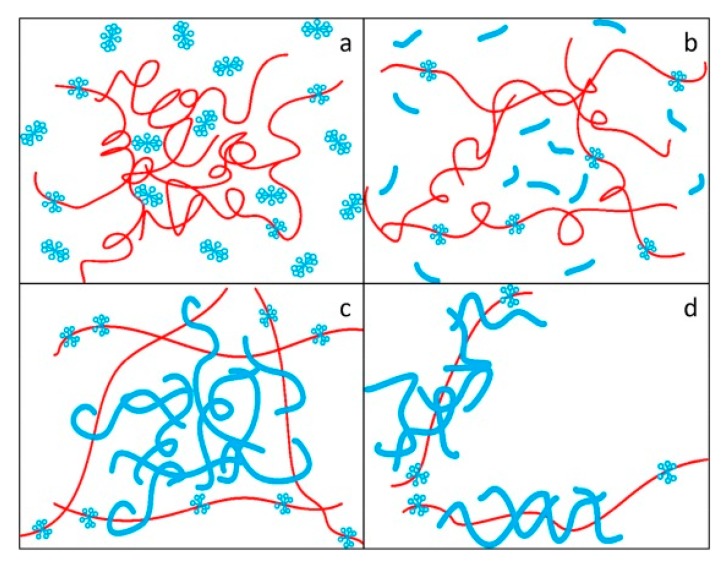
Formation and failure of a complex network structure model: (**a**)dispersed polymer molecules and surfactant molecules; (**b**) stretching polymer molecules and rodlike micelles; (**c**) the network structure model of polymer molecules and polymer molecules; (**d**) the broken network structure model [45].

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
