# Peer review of "Research Progress on the Collaborative Drag Reduction Effect of Polymers and Surfactants"

_materials, 2020, doi:10.3390/ma13020444_

Round 1

Reviewer 1 Report

The article is a brief review of studies on the use of additives to reduce drag.
The text of the article describes hypotheses explaining the Toms effect and the effect of reducing the degradation of the "anti-turbulent" properties of polymer additives.

The title of the article indicates that the article is devoted to progress in studies of the collaborative action of polymers and surfactants to reduce drag in liquids. But the article does not contain enough information on studies of the effect of surfactant additives on hydraulic drag.

The authors write that studies of the mechanism of drag reduction began only in 1980, which is not entirely true.

The text of the article did not discusses the mechanism of reducing the drag when surfactants are added to the liquid.
Figure 2 does not indicate what is shown in the pictures a) b) and c).

The article is devoted to additives that reduce hydraulic drag. But, from the text, it becomes clear that additives are meant polymers and surfactants. Also in the text, much attention is paid to the effects of the combined addition of surfactants and polymers to a liquid. But a significant part of the article is devoted to the effect of polymer additives on hydraulic resistance and nothing is said about the mechanism of action of surfactant additives.
But surfactants also reduce hydraulic resistance and it would be nice to also consider the mechanism of their action.

The text should give an explanation of what the following abbreviations mean:

PEO, PAM, CMC, SDS, PAA, RPM, CAC, HGA, HDBAC and etc.

Author Response

Response to Reviewer 1 Comments

Thank you very much for your comments and suggestions concerning our manuscript entitled “Research Progress of Collaborative Drag Reduction Effect of Polymer and Surfactant” (materials-692868). Those suggestions are of great help for revising and improving the paper. We have studied those comments carefully and tried our best to revise the paper. I list the main corrections requested by the reviewers as the following.

Point 1: The authors write that studies of the mechanism of drag reduction began only in 1980, which is not entirely true. 

Response 1: Thanks for the comment. We revised "1980s" to "1948". Mysels, an American scholar, found that the phenomenon of turbulent drag reduction occurs when aluminum bisfatty acid is dissolved into gasoline. After the rheology conference in 1948, the research was officially started. So we consider this point in time as the beginning.

Point 2: The text of the article did not discusses the mechanism of reducing the drag when surfactants are added to the liquid.

Response 2: The mechanism of drag reduction of surfactants is explained from the microscopic point of view in section 1.4. We supplemented the relationship between surfactant micelle structure and drag reduction mechanism.

Point 3: Figure 2 does not indicate what is shown in the pictures a) b) and c).

Response 3: In figure 2, figure 2(a) is the polymer-free solution, figure 2(b) is the poly epoxyethane 1.5×10−3 weight percentage solution, and figure 2(c) is the hydroxypropyl cellulose 0.08 weight percentage solution. The flow field in Figure 2 (b) is affected by poly epoxyethane, and the vortex structure in the flow became long and thin.

Point 4: The article is devoted to additives that reduce hydraulic drag. But, from the text, it becomes clear that additives are meant polymers and surfactants. Also in the text, much attention is paid to the effects of the combined addition of surfactants and polymers to a liquid. But a significant part of the article is devoted to the effect of polymer additives on hydraulic resistance and nothing is said about the mechanism of action of surfactant additives.

But surfactants also reduce hydraulic resistance and it would be nice to also consider the mechanism of their action.

Response 4: From the physical point of view of turbulence, the explanation of turbulence drag reduction mechanism is applicable to both polymer drag reduction agents and surfactant drag reduction agents. Therefore, there is no distinction between the types of chemical drag reduction agents in this aspect, although some explanations of turbulence drag reduction mechanism are aimed at polymer. In order to explain the mechanism of drag reduction from the point of view of the internal microstructure of drag reduction agent solution, it is necessary to elaborate the difference between polymer drag reduction agent and surfactant drag reduction agent. In section 1.4, the mechanism of drag reduction of surfactants is explained from the microscopic point of view.

Point 5: The text should give an explanation of what the following abbreviations mean:

PEO, PAM, CMC, SDS, PAA, RPM, CAC, HGA, HDBAC and etc.

Response 5: Thank you for your suggestions. All abbreviations are given in full in the text.

Reviewer 2 Report

This peer-reviewed paper is a review of studies on the use of polymer additives and surfactants as drag reduction agents. The topic of the review is important and its content is useful. However, the paper needs to be improved before it can be accepted for publication.

It seems that the authors use the original figures of the cited publications.
1.1. Are copyrights properly respected? I expect it can be a problem.
1.2. The use of the original figures is a bad choice in this case. None of the figures is discussed in detail. They should be adopted to the text of the review. Structure of the review. The efficiency of the use of polymer additives, surfactants or their mixtures as drag reduction agents will depend on the nature of the solution. The author should separate studies on organic liquids and aqueous solutions. The review is readable but there are many grammatical and semantic errors.
3.1. 87-88: „With the emergence of advanced measuring instruments, researchers found that the turbulent bottom layer close to the pipe wall is not a simple laminar flow, but a longitudinal velocity fluctuation” The turbulent layer cannot be a laminar one.
3.2. 96-97: “This may seem reasonable at first, but it is not consistent with the measured results of turbulence fluctuation intensity.” The meaning of the sentence is not explained.
3.3. 185-187: “As shown in Figure 5, (a) is the formation and evolution of spherical micelle, and (b-d) is the gradual transformation of spherical micelles into wormlike micelle. The morphology changed from spherical micelle to insect micelle.” Worms are not insects!
3.4. 166-173: “Yang [34] used the concept of elastic shear stress or Reynolds shear stress deficit to derive the velocity distribution and friction coefficient equations in turbulent drag reduction flow. The initial Reynolds number of drag reduction depends on the type of polymer and its concentration. The optimum polymer concentration for the maximum drag reduction ratio depends only on the type of polymer. Eskin [35] adjusted the modeling method of Yang, and considered the influence of pipe diameter on pipe flow drag reduction. To verify this adjustment, the calculation results of the drag reduction model based on Yang method are compared with the experimental data of Virk [8], and the data conform to the experimental data of Virk.” I do not understand how this affects degradation. Temperature should be given in Celsius or Kelvin not Fahrenheit (164: “at 100 ° F when”)

Author Response

Response to Reviewer 2 Comments

Thank you very much for your comments and suggestions concerning our manuscript entitled “Research Progress of Collaborative Drag Reduction Effect of Polymer and Surfactant” (materials-692868). Those suggestions are of great help for revising and improving the paper. We have studied those comments carefully and tried our best to revise the paper. I list the main corrections requested by the reviewers as the following.

Point 1: 1.1. Are copyrights properly respected? I expect it can be a problem.

Response 1: All the pictures in the review have been copyrighted, and the electronic documents of copyright license have been submitted to the editor when the manuscript is submitted for the first time.

Point 2: 1.2. The use of the original figures is a bad choice in this case. None of the figures is discussed in detail. They should be adopted to the text of the review. Structure of the review. The efficiency of the use of polymer additives, surfactants or their mixtures as drag reduction agents will depend on the nature of the solution. The author should separate studies on organic liquids and aqueous solutions. The review is readable but there are many grammatical and semantic errors.

Response 2: Thanks for the comments. We added some comments on pictures 1, 2 and 3. In particular, in Figure 3, the drag reduction mechanism of surfactants is described. For the type of solution that the drag reducer acts on, whenever there is a specific drag reducer name, it is pointed out that their drag reducer solution is an organic solution or an aqueous solution.

Point 3: 3.1. 87-88: “With the emergence of advanced measuring instruments, researchers found that the turbulent bottom layer close to the pipe wall is not a simple laminar flow, but a longitudinal velocity fluctuation” The turbulent layer cannot be a laminar one.

Response 3: It has been revised. We changed “the turbulent layer” to “the laminar layer”.

Point 4: 3.2. 96-97: “This may seem reasonable at first, but it is not consistent with the measured results of turbulence fluctuation intensity.” The meaning of the sentence is not explained.

Response 4: Thanks for the comment. We added the explanation of this sentence. “With the development of measurement technology, such as PIV, it is found that the structure of turbulence itself has changed, especially vortex. Therefore, the reason of turbulence drag reduction is not only velocity fluctuation.” At the same time put this sentence in the back. “In the downstream of the area where drag reducer is added, the hairpin vortex along the shear layer is significantly reduced, and the energy loss caused by the corresponding vortex is reduced [16, 17].”

Point 5: 3.3. 185-187: “As shown in Figure 5, (a) is the formation and evolution of spherical micelle, and (b-d) is the gradual transformation of spherical micelles into wormlike micelle. The morphology changed from spherical micelle to insect micelle.” Worms are not insects!

Response 5: We changed “insect” to “wormlike”.

Point 6: 3.4. 166-173: “Yang [34] used the concept of elastic shear stress or Reynolds shear stress deficit to derive the velocity distribution and friction coefficient equations in turbulent drag reduction flow. The initial Reynolds number of drag reduction depends on the type of polymer and its concentration. The optimum polymer concentration for the maximum drag reduction ratio depends only on the type of polymer. Eskin [35] adjusted the modeling method of Yang, and considered the influence of pipe diameter on pipe flow drag reduction. To verify this adjustment, the calculation results of the drag reduction model based on Yang method are compared with the experimental data of Virk [8], and the data conform to the experimental data of Virk.” I do not understand how this affects degradation.

Response 6: Thank you for your advice. Pipe diameter mainly affects the drag reduction rate. We removed the part that was not closely related to degradation. It has been changed into a literature on the concentration and degradation of drag reducers.

Point 7: Temperature should be given in Celsius or Kelvin not Fahrenheit (164: “at 100 ° F when”)

Response 7: Thanks for the comment. We change the temperature unit from Fahrenheit to Celsius.

Round 2

Reviewer 1 Report

Everything is fine.

Reviewer 2 Report

The paper can be recommended for publication. I do not have other important comments.